# Replicating the Disease framing problem during the 2020 COVID-19 pandemic: A study of stress, worry, trust, and choice under risk

Nikolay R. Rachev[1]*, Hyemin Han[2], David Lacko[3], Rebekah Gelpí[4], Yuki Yamada[5], Andreas Lieberoth[6]

1 Department of General, Experimental, Developmental, and Health Psychology, Sofia University St. Kliment Ohridski, Sofia, Bulgaria, 2 Educational Psychology Program, University of Alabama, Tuscaloosa, Alabama, United States of America, 3 Interdisciplinary Research Team on Internet and Society, Faculty of Social Studies, Masaryk University, Brno, Czechia, 4 Department of Psychology, University of Toronto, Toronto, Ontario, Canada, 5 Faculty of Arts and Science, Kyushu University, Fukuoka, Japan, 6 Danish School of Education, Aarhus University, Aarhus, Denmark

☯ These authors contributed equally to this work.
* nrrachev@phls.uni-sofia.bg

**Data Availability Statement:** All data files and code are available from the Open Science Framework database, http://doi.org/10.17605/OSF.IO/GQXCH.

## Abstract

In the risky-choice framing effect, different wording of the same options leads to predictably different choices. In a large-scale survey conducted from March to May 2020 and including 88,181 participants from 47 countries, we investigated how stress, concerns, and trust moderated the effect in the Disease problem, a prominent framing problem highly evocative of the COVID-19 pandemic. As predicted by the appraisal-tendency framework, risk aversion and the framing effect in our study were larger than under typical circumstances. Furthermore, perceived stress and concerns over coronavirus were positively associated with the framing effect. Contrary to predictions, however, they were not related to risk aversion. Trust in the government's efforts to handle the coronavirus was associated with neither risk aversion nor the framing effect. The proportion of risky choices and the framing effect varied substantially across nations. Additional exploratory analyses showed that the framing effect was unrelated to reported compliance with safety measures, suggesting, along with similar findings during the pandemic and beyond, that the effectiveness of framing manipulations in public messages might be limited. Theoretical and practical implications of these findings are discussed, along with directions for further investigations.

## Introduction

"Imagine that your country is preparing for the outbreak of an unusual disease, which is expected to kill many people." This is, roughly, the first sentence of the Disease problem, a classic problem illustrating the framing effect, whereby different presentations of the same problem lead to systematically different choices [1]. Originally proposed as a hypothetical scenario, it sounds highly evocative of the COVID-19 pandemic. We were interested in how people around the world would respond to this famous problem when they are genuinely facing a

**Funding:** The authors disclose receipt of the following financial support for the publication of this article after the decision of acceptance: Japan Society for the Promotion of Science (https://www.jsps.go.jp/english/) KAKENHI Grant Numbers JP20H04581 (Y.Y.). The funders had no role in study design, data collection and analysis, decision to publish, or preparation of the manuscript.

**Competing interests:** The authors have declared that no competing interests exist.

disease that has changed, and perhaps even threatened, their lives. In particular, we used the naturalistic setting provided by the COVID-19 pandemic to test predictions from the appraisal-tendency theory [2–4] regarding the moderating role of stress, worry, and trust on the framing effect. Our large-scale dataset from various countries around the world, within the COVIDiSTRESS global survey [5, 6], allowed us to also investigate the cross-national variability of the effects of interest. More relevant to pandemic management, we explored whether choice under risk relates to compliance with the healthcare guidelines.

## The framing effect and the disease problem

The framing effect, in the context of risky choices, refers to the shift of preferences depending on superficial changes in the wording of otherwise equivalent choice options, in violation of the rational principle of descriptive invariance [1, 7, 8]. Specifically, people tend to be risk-averse when presented with potential gains but tend to be risk-seeking when the same options are presented as potential losses.

The single most popular framing task is the Disease problem (DP) [1]. In the gain frame, it reads as follows:

> Imagine that the U.S. is preparing for the outbreak of an unusual Asian disease, which is expected to kill 600 people. Two alternative programs to combat the disease have been proposed. Assume that the exact scientific estimate of the consequences of the programs are as follows:
>
> · If Program A is adopted, 200 people will be saved.
>
> · If Program B is adopted, there is 1/3 probability that 600 people will be saved, and 2/3 probability that no people will be saved
>
> Which of the two programs would you favor?
>
> In the loss frame, the same choice options are phrased as follows:
>
> · If Program C is adopted 400 people will die.
>
> · If Program D is adopted there is 1/3 probability that nobody will die, and 2/3 probability that 600 people will die.

The vast majority of participants in the original study chose the sure Program A in the gain frame (72%) and the risky Program D in the loss frame (78%) [1]. Multiple subsequent replications [9] and meta-analyses [10–12] showed that the effect is robust albeit smaller in size than in the original study [1].

The DP has received researchers' attention more than any other framing task [12], serving as a critical test of prominent accounts of the framing effect. Multiple accounts have been proposed which differ widely in the hypothesized underlying mechanisms, including whether and how people violate rational principles. For instance, people might violate descriptive invariance by following the psychophysical principles of a reference point and diminished sensitivity [1, 7] or by extracting the gist as opposed to verbatim mental representation of the problem [13]. It is also possible that people do not violate rational principles because the two frames elicit different inferences about information that is not explicitly provided; that is, the frames are not informationally equivalent [14, 15]. While these major accounts are fundamentally different, none of them has focused on emotions, although the potential of emotions to modulate the framing effect has been suggested ever since the effect has been demonstrated [1].

More recently, the role of emotions has received more attention. For instance, susceptibility to framing was associated with amygdala activity while reduced susceptibility to framing was associated with prefrontal cortex activity in a financial decision-making task [16]. As amygdala activity is linked to emotions while the prefrontal cortex is linked to executive functioning, these findings suggest that emotional engagement enhances the framing effect while deliberate thinking reduces it. Yet, further evidence has been mixed and inconsistent. Relative to a control condition, focusing on emotions sometimes selectively enhanced the framing effect among males but not females [17], and sometimes did not alter the size of the effect [18, 19]. More specifically, positive affect either eliminated the framing effect [20, 21] or increased the effect relative to a neutral condition [18] or led to an effect of the same size as the neutral condition but larger than under negative affect [19]. Similarly, negative affect either attenuated the effect [21, 22] or was unrelated to it [18–20]. Resolving these conflicting findings might require going beyond positive versus negative valence in studying the role of emotions in the framing effect.

## Appraisal-tendency framework, risk seeking, and the framing effect

The appraisal-tendency framework (ATF) [2, 3] explicitly takes into account cognitive dimensions of emotions, in addition to their valence. According to ATF, emotions that share the same valence can be associated with different cognitive appraisals which have distinct carry-over effects on subsequent judgments and decisions. In particular, negative emotions related to anxiety, such as distress and fear, are accompanied by appraisals of uncertainty and high situational control [23] and will thus lead to risk avoidance. By contrast, negative emotions related to aversion, such as anger and hostility, are associated with an appraisal of certainty and high other-person control [23] and will lead to risk-seeking [2]. Anxiety also enhances sensitivity to contextual cues, such as frames, while aversion enhances behavior motivated more by personal dispositions, thus drawing attention away from frames [4]. Therefore, ATF predicts that anxiety increases while aversion decreases the framing effect. Previous research [4, 24] has largely confirmed these predictions, with the effects more apparent in the DP than in an investment problem [4].

Going beyond laboratory studies, participants in a nationally representative sample of US citizens studied shortly after September 11, 2001 [25] assigned a lower probability to terrorism-related and routine risks when induced with anger rather than fear. In a follow-up study [26], the anger versus fear induction had a differential effect not only on judgments of future risks (albeit smaller in magnitude than those found in [25]) but also on the reevaluation and recall of past judgments. These findings support ATF's prediction that fear and anxiety enhance risk avoidance while anger enhances risk-seeking. However, these studies have not tested the moderating role of these emotions on the framing effect.

We aimed to test ATF's predictions using the DP in two ways. First, we compared our data to the open-access data provided by the Many Labs Replication Project [9], an international effort aiming to replicate 13 well-known effects in psychology, including the framing effect in the DP. That project collected data from 36 different samples ($N = 6,344$) in ten different countries (but dominated by US samples) both in lab settings and online. It thus provided a robust estimate of the size of the framing effect, Cohen's $d = 0.62$, which is very close to estimates in meta-analyses [10, 11]. We compared the proportions of safe choices and the size of the framing effect in this reference dataset to those in our data collected under the early months of the COVID-19 pandemic, presumably under more stressful circumstances. As the Many Labs have not measured stress level, we used US data from 2009 [27] as a rough estimate of the stress level in a period close to their data collection.

The above approach tests ATF's predictions indirectly: it can potentially show that stress, risk aversion, and the framing effect have increased relative to a pre-pandemic period but it does not directly test whether the increases are associated with one another. Neither can it rule out the possibility that factors other than the pandemic have driven the increases. Therefore, as a stronger and more direct test of ATF, we also tested the hypothesized associations within our dataset. Specifically, we predicted that higher distress and anxiety would be associated with a larger proportion of safe choices and a larger framing effect.

We have not measured anger or hostility, so we were not able to directly test ATF's predictions regarding these emotions. We have, however, measured trust. Although not an emotion itself, trust shares important features with anger and hostility, in particular the appraisal of other-person control and, in low levels of trust, a shared negative valence. In prior research, anger was negatively associated with judgments of trust and positively associated with judgments of distrust [28]. Incidental anger also lowered unrelated trust judgments [29]. In an imaginary negotiation, distrust also mediated the impact of anger on non-cooperative behavioral intentions [28]. Distrust, but not trust, is also associated with brain areas related to intense negative emotions [30]. Anger and distrust can thus potentiate each other in valence-congruent ways to influence behavior. Accordingly, we hypothesized that lower trust would be associated with a larger proportion of risky choices and a reduced or reversed framing effect by virtue of appraisal mechanisms similar to anger and hostility.

## Cross-national variations in risk attitudes and susceptibility to framing

The attitude of risk aversion under gain framing and risk-seeking under loss framing seems to be robust across many nations [31]. The degree of risk aversion, however, varies cross-nationally [31–33]. Accordingly, we expected that risk aversion and the framing effect will differ across countries.

What predicts cross-national differences is less clear. In several studies, participants from individualistic Western countries were more risk-averse than their counterparts from collectivistic East Asian countries in financial choices [31, 34, 35]. This association was not replicated, however, in a large-scale study [36] which found instead a stronger preference for safe choices among participants from countries with higher versus lower GDP per capita (see also [31, 32]). Participants from countries with a higher GDP per capita were also more risk-averse under a gain frame and more risk-seeking under a loss frame [31]. Since findings accounted for by cultural differences are also consistent with GDP differences [36], we explored whether GDP per capita would be associated with risk preferences on the DP, similarly to findings using monetary outcomes.

## Risky-choice framing and compliance with safety guidelines

Attitudes towards risk and related social motives are important predictors of safety measures during the COVID-19 pandemic. For instance, individuals who are more risk-tolerant [37] or are more prone to expose others at risk for their own benefit in an incentivized game [38] are also less willing to comply with safety guidelines. Accordingly, we investigated whether choosing the safe option on the DP would be related to higher compliance.

We also explored if compliance would be associated with the framing effect. If the framing manipulation has any effect beyond the DP, then presenting the information in a gain rather than a loss frame would encourage safety choices. A positive finding would be in line with the literature citing framing manipulations as a powerful tool for directing behavior [e.g., 39]. A positive finding would also suggest that framing of public messages might favorably

influence behavioral outcomes at scale, thus providing an easy and effective solution to the central challenge to policy-makers of presenting information to the public during crises [5, 40, 41].

Yet, there are also many reasons why the choice on the DP might not transfer to everyday choices. While the DP is highly evocative of the pandemic, the choice it prompts is quite unusual and unlike the more ordinary choices related to compliance in many respects. The former is a single choice between two options affecting a fixed number of other people with probabilities and values known in advance. The latter is a continuous choice among a large combination of behaviors which affects mostly oneself but also an unknown number of other people, with no numerical values attached but with a clear message regarding the socially desirable choice. Since the framing manipulation in the DP did not directly target choices concerning compliance, it might have a limited role in influencing these behaviors. Therefore, we investigated the associations with compliance only exploratively.

### Preregistered hypotheses

Our preregistered hypotheses were related primarily to ATF's predictions regarding the moderation role of emotions and, second, to cross-national differences in framing. Hypotheses 1a and 1b concerned comparisons between our dataset and the Many Labs dataset [9]. We hypothesized that, if overall perceived distress was higher than typical, then,

(Hypothesis 1a) the proportion of "safe" choices on the Disease problem, regardless of frame, would be higher than typical (as compared to [9]), and

(Hypothesis 1b) the framing effect on the Disease problem would be larger than typical, as represented by [9], in terms of Cohen's d.

Hypotheses 2a—2f tested ATF predictions within our study. We anticipated that higher levels of overall distress would be associated with:

(Hypothesis 2a) a larger proportion of "safe" choices in the Disease problem regardless of frame, and

(Hypothesis 2b) a larger framing effect.

We also expected that higher levels of coronavirus concerns would be associated with:

(Hypothesis 2c) a larger proportion of safe choices on the Disease problem regardless of frame, and

(Hypothesis 2d) a larger framing effect.

Finally, we hypothesized that lower trust in the country's government and health system will be associated with:

(Hypothesis 2e) a larger proportion of risky choices on the Disease problem regardless of frame, and

(Hypothesis 2f) a smaller or reversed framing effect.

Concerning cross-national differences, we anticipated that:

(Hypothesis 3a) the proportion of risky choices would vary across countries, and

(Hypothesis 3b) the size of the framing effect would vary across countries.

### Exploratory analyses

We preregistered three exploratory analyses for questions we found practically important but for which we could not find sufficient theoretical background to draw hypotheses from. We first explored whether GDP would predict choice and moderate the framing effect, as found in financial problems [31, 32, 36]. Second, we explored whether compliance with health guidelines was associated with risk-taking and whether compliance was associated with the framing effect. Finally, we investigated whether self-reported familiarity with the DP was associated with the proportion of "safe" choices and with the framing effect since familiarity might significantly affect participants' responses. We also performed non-preregistered exploratory analyses as a follow-up to unexpected findings, as a more fine-grained exploration, or as a robustness check (most of these in response to reviewers' suggestions).

## Materials and methods

### Overview

The COVIDiSTRESS global survey [5, 6] (available at https://osf.io/mhszp/) was an open science project conducted in collaboration between about 150 researchers around the world aiming to map the early psychological and behavioral responses to the COVID-19 pandemic across countries. It was translated into 47 languages and dialects, and was distributed via Qualtrics between March 30th and May 30th, 2020. Participants were recruited via social media. They provided written informed consent before accessing the survey.

The project was evaluated by the Institutional Review Board at Aarhus University. Given the time-sensitive nature of data collection, an initial waiver (case number: 2019-616-000009) was obtained on March 23 from the IRB to proceed with data collection, followed by a formal approval on June 10 (case number: 2020–0066175).

The survey consisted of two parts. The first part included questions regarding the direct effects of the pandemic on the respondents' everyday life (e.g., isolation status and first-hand experience), perceived stress and loneliness, personality traits, and daily behaviors including compliance with preventive measures. The second part consisted of more specific items related to coronavirus concerns, coping, and social provisions. The full list of measures is reported elsewhere [5]. Here, we provide details only for the measures relevant to the present research.

### Preregistration

We preregistered a research protocol on the Open Science Framework [42] which included our hypotheses, design plan, sampling plan, variables, and exploratory analyses. This was made after the data were collected, but before analyzing the data, the only exclusion being a preliminary analysis of our Hypothesis 1a. The preliminary analysis was performed to discuss the analysis code among the researchers. It was conducted with a dataset that was not pre-processed according to our preregistered data exclusion criteria. We adhered to the preregistered protocol in implementing and analyses. The few instances where we deviated from the preregistered protocol are reported and justified in, S1.1 Table in S1 File.

### Participants

The full COVIDiSTRESS dataset [6] contained 173,426 responses from 179 countries over the globe. We used the cleaned version of the COVIDiSTRESS dataset [6] (see https://osf.io/e2y7w/ for the cleaning procedures) consisting of 125,306 respondents who met the inclusion criteria (18 years of age and older and gave informed consent). From this sample, we excluded: 1) participants who filled in the survey in less than one-tenth of the estimated time (i.e., < 2

min and 12 sec, $n$ = 1,288); 2) participants with missing data on any of the variables of interest ($n$ = 31,171); 3) participants from countries with less than 100 participants (needed for the purposes of the measurement invariance analysis and multilevel modeling; $n$ = 1,211); and 4) multivariate outliers based on Mahalanobis distance (i.e. participants with a $p <$ .001 in the chi-square test; $n$ = 3,455). After excluding multivariate outliers, all countries had at least 100 participants except Russia which had 97 participants. We decided to keep these observations in the dataset.

Our research sample thus consisted of 88,181 participants (72.50% female, 26.32% male, 1.01% other/would rather not say, 0.17% no data) aged between reported 18 to 110 years ($M$ = 38.84, $SD$ = 13.79), from 47 countries. The descriptive statistics for each country are reported in Table 1.

We examined whether exclusions led to selection bias by comparing the gender and age of included versus excluded participants in our dataset. Among those who reported their gender, 73.37% were female among the included versus 72.73% among the excluded participants. Given the sensitivity of significance testing to sample size, the difference in these proportions was statistically significant, $Z$ = 2.30, $p <$ .05 but of negligible size, odds ratio = .97. Similarly, the difference between age for included ($M$ = 38.84 years, $SD$ = 13.79) versus excluded ($M$ = 40.12, $SD$ = 14.75 years) participants was statistically significant, $t(65,711)$ = 14.33, $p <$ .001 but very small, Cohen's $d$ = .11. Thus, we concluded that data pre-processing did not lead to selection bias.

We also compared the gender composition of our dataset versus the Many Labs dataset [9] that we used for a reference. Among those who reported their gender in our study, 63,935 participants (73.37%) were female while 23,205 (26.63%) were male. In [9], 2,052 (69.77%) were female while 889 (30.23%) were male. The difference in proportions was statistically significant $Z$ = 4.33, $p <$ .001, but of negligible size, odds ratio = 1.19.

## Measures

**Disease problem.**   Participants were asked to imagine that their own country (as opposed to the U.S.) was facing an unusual disease (with the word "Asian" omitted). Besides these two differences, the English version was identical to the original [1, see Introduction for full text]. Frame (gain vs. loss) was manipulated between participants. After making their choice, participants disclosed their familiarity with the DP by answering the following item: "The previous question has been used in other research too. Did you recognize it, and if so, did you remember what the original study was about?" Participants were provided with three response options, "yes," "no," and "not sure."

**Perceived stress scale (PSS-10).**   PSS-10 [43, 44] is a 10-item self-report scale designed to measure the extent to which everyday situations are perceived as stressful. In the present survey, participants were asked about their feelings and thoughts during the last month as opposed to last week (as in the original scale). A sample item is "In the last week, how often have you felt nervous and 'stressed'?" Participants responded on a 5-point scale (1 = "never"; 5 = "very often").

**Coronavirus concerns.**   Participants indicated how much they agreed, on a 6-point Likert scale (1 = "strongly disagree"; 6 = "strongly agree") that they were concerned about the consequences of the coronavirus for (1) themselves, (2) their family, (3) their close friends, (4) their country, and (5) other countries across the globe.

**Trust in the country's institutions.**   Participants indicated the extent to which they personally trusted, on a 11-point Likert scale (0 = "not at all"; 10 = "completely"), each of the following institutions: (1) their country's Parliament/government, (2) their country's police,

**Table 1. Participants: Sample size, age (mean and standard deviation), and distribution by gender across countries included in the analyses.**

| Country | N | Age | | Gender (%) | | |
|---|---|---|---|---|---|---|
| | | *Mean* | *SD* | Female | Male | Other |
| **Argentina** | 3592 | 39.24 | 14.58 | 83.41 | 15.56 | 0.97 |
| **Australia** | 237 | 42.37 | 13.49 | 75.53 | 23.21 | 1.27 |
| **Austria** | 220 | 38.32 | 11.68 | 68.64 | 30 | 0.91 |
| **Bangladesh** | 226 | 27.99 | 6.39 | 43.81 | 55.31 | 0.88 |
| **Belgium** | 476 | 36.45 | 12.54 | 57.56 | 41.6 | 0.42 |
| **Bosnia and Herzegovina** | 761 | 35.83 | 11.15 | 74.64 | 24.57 | 0.39 |
| **Brazil** | 476 | 34.5 | 13.12 | 71.85 | 27.52 | 0.42 |
| **Bulgaria** | 2877 | 40.37 | 13.47 | 80.15 | 18.35 | 1.39 |
| **Canada** | 353 | 40.92 | 14.03 | 66.57 | 30.88 | 2.27 |
| **Colombia** | 128 | 32.91 | 11.4 | 60.16 | 38.28 | 0.78 |
| **Croatia** | 2129 | 34.71 | 11.97 | 79.38 | 20.06 | 0.28 |
| **Czech Republic** | 1417 | 32.68 | 11.33 | 78.9 | 20.54 | 0.56 |
| **Denmark** | 8463 | 41.92 | 14.01 | 78.66 | 20.83 | 0.35 |
| **Finland** | 18444 | 43 | 13.89 | 81.33 | 16.74 | 1.77 |
| **France** | 9777 | 32.69 | 12.34 | 51.56 | 46.91 | 1.32 |
| **Germany** | 1073 | 36.29 | 11.81 | 69.25 | 29.08 | 1.49 |
| **Greece** | 472 | 41.47 | 11.77 | 77.12 | 22.67 | 0.21 |
| **Hungary** | 890 | 47.35 | 14.77 | 67.19 | 32.02 | 0.34 |
| **Indonesia** | 1036 | 30.28 | 8.95 | 67.76 | 30.89 | 1.16 |
| **Ireland** | 148 | 39.34 | 9.94 | 79.73 | 18.92 | 1.35 |
| **Italy** | 1188 | 42.54 | 14.99 | 76.43 | 22.56 | 0.76 |
| **Japan** | 3871 | 44.77 | 11.26 | 43.55 | 55.49 | 0.93 |
| **Korea, South** | 330 | 38.55 | 10.17 | 47.58 | 51.21 | 0.91 |
| **Kosovo** | 1319 | 28.09 | 9.06 | 63.38 | 35.48 | 0.91 |
| **Lithuania** | 6260 | 38.03 | 12.02 | 75.42 | 24.01 | 0.51 |
| **Malaysia** | 399 | 36.21 | 13.95 | 74.69 | 24.06 | 1 |
| **Mexico** | 6406 | 36.42 | 13.28 | 71.85 | 27.29 | 0.53 |
| **Netherlands** | 1101 | 44.78 | 14.49 | 73.75 | 25.25 | 0.73 |
| **New Zealand** | 100 | 39.54 | 12.25 | 88 | 11 | 1 |
| **Norway** | 132 | 39.45 | 10.63 | 69.7 | 29.55 | 0.76 |
| **Pakistan** | 193 | 26.64 | 7.65 | 71.5 | 27.46 | 0 |
| **Panama** | 448 | 37.4 | 14.14 | 72.32 | 26.56 | 0.45 |
| **Philippines** | 409 | 25.18 | 10.4 | 67.48 | 31.78 | 0.73 |
| **Poland** | 2083 | 31.15 | 7.57 | 86.94 | 12.34 | 0.72 |
| **Portugal** | 767 | 32.55 | 12.81 | 86.44 | 13.04 | 0.26 |
| **Romania** | 182 | 33.66 | 8.97 | 73.63 | 26.37 | 0 |
| **Russia** | 97 | 32.76 | 12.23 | 73.2 | 26.8 | 0 |
| **Serbia** | 155 | 38.21 | 12.52 | 63.23 | 36.13 | 0.65 |
| **Slovakia** | 636 | 41.51 | 12.92 | 77.99 | 21.54 | 0.47 |
| **Spain** | 418 | 38.07 | 15.15 | 71.05 | 28.71 | 0 |
| **Sweden** | 2296 | 45.82 | 12 | 75.26 | 23.65 | 0.91 |
| **Switzerland** | 921 | 42.83 | 17.41 | 61.24 | 37.89 | 0.33 |
| **Taiwan** | 1669 | 33.63 | 11.72 | 67.94 | 29.9 | 2.1 |
| **Turkey** | 735 | 32.39 | 11.19 | 74.29 | 24.9 | 0.68 |
| **United Kingdom** | 1082 | 39.27 | 12.69 | 76.43 | 22.92 | 0.55 |
| **United States** | 1652 | 42.35 | 14.56 | 77.12 | 21.43 | 1.39 |

*(Continued)*

**Table 1.** (Continued)

| Country | N | Age | | Gender (%) | | |
|---|---|---|---|---|---|---|
| | | *Mean* | *SD* | Female | Male | Other |
| **Vietnam** | 137 | 24.77 | 6.63 | 68.61 | 30.66 | 0.73 |

*Note.* N = Number of participants.

(3) their country's civil service, (4) their country's health system, (5) The World Health Organization (WHO), (6) their country's government's effort to handle the Coronavirus.

**Compliance with local prevention guidelines.** Participants indicated their agreement, on a 6-point Likert scale (1 = strongly disagree, 6 = strongly agree) with the statement "I have done everything I could possibly do as an individual, to reduce the spread of coronavirus".

## Statistical analyses

**Measurement invariance test and measurement alignment.** Prior to analyses, we tested for measurement invariance of PSS-10, Coronavirus Concerns, and Trust in the Country's Institutions. The analyses were performed in *R* version 4.0.2 [45], using packages *lavaan* version 0.6.7 [46] and *semTools* version 0.5–3 [47]. Using a multi-group confirmatory factor analysis, we considered three models: one assuming a two-factor structure for PSS-10 (positive and negative, with the latter consisting of reversed items [48, 49]) and one-factor structure for Coronavirus Concerns and Trust across all countries (configural invariance), a second model with factor loadings and latent correlations constrained to be equal (metric invariance), and a third model with equal items' intercepts in all groups (scalar invariance). We compared the configural invariance model with the metric invariance model, and then the metric invariance model with the scalar invariance model [50]. To determine whether measurement invariance was achieved, we used -.01 change in CFI, +.015 in RMSEA, and +.030 in SRMR, for metric invariance, and -.01 change in CFI, +.015 in RMSEA, and +.015 in SRMR, for scalar invariance [51]. Scalar measurement invariance, which is necessary for comparing means across various countries, was not achieved in all cases (see Table 2).

The internal consistency of PSS-10 was good, Cronbach's α = .88. However, the two-factor as well as one-factor structure, which was used due to the multigroup measurement alignment (see next paragraph), showed scalar non-invariance. The internal consistency of the Coronavirus Concerns scale was good, α = .82, but its configural model showed unsatisfactory fit indices. Following [5], we used only its first three items which also had good internal consistency,

**Table 2. Measurement invariance.**

| Scale | Level | RMSEA | SRMR | CFI | TLI | ΔRMSEA | ΔSRMR | ΔCFI | ΔTLI |
|---|---|---|---|---|---|---|---|---|---|
| **PSS-10 (2-factor solution)** | Configural | .05 | .03 | .97 | .96 | - | - | - | - |
| | Metric | .05 | .04 | .96 | .96 | .00 | .02 | -.01 | -.00 |
| | Scalar | .08 | .07 | .88 | .89 | .03 | .03 | -.08 | -.07 |
| **PSS-10 (1-factor solution)** | Configural | .08 | .04 | .92 | .90 | - | - | - | - |
| | Metric | .07 | .06 | .94 | .93 | -.01 | .02 | .02 | .03 |
| | Scalar | .11 | .09 | .79 | .82 | .04 | .03 | -.15 | -.11 |
| **Concerns (5 items)** | Configural | .24 | .10 | .56 | .11 | - | - | - | - |
| **Concerns (3 items)** | Configural | .00 | .00 | 1.0 | 1.0 | - | - | - | - |
| | Metric | .08 | .02 | .98 | .96 | .08 | .02 | -.02 | -.04 |
| **Trust** | Configural | .12 | .04 | .90 | .82 | - | - | - | - |

$\alpha$ = .85 but showed metric non-invariance. The Trust in the Country's Institutions scale also had good internal consistency, $\alpha$ = .89 but yielded configural non-invariance. Following [5], we only used one item for subsequent analyses, namely trust in the government's effort to handle coronavirus, which is directly related to the context of the pandemic.

In addition, the modification indices did not indicate any constraints to be freed to improve the model fit in all cases, which impeded testing for partial invariance [52]. Given that even partial invariance is sometimes difficult to achieve when comparing numerous countries [53–55], we addressed the non-invariance issue using the alignment method implemented in *R* package *sirt* [56] to adjust factor loadings and intercepts across countries. Because measurement alignment is currently available for one-factor models [56], we implemented a one-factor model for each scale. We obtained an $R^2$ value of .97 for loadings and 1.00 for intercepts in the case of PSS-10, and an $R^2$ .99 for loadings and 1.00 for intercepts in the case of the Coronavirus Concerns scale. These values suggest that the most of non-invariance was absorbed with the factor loading and intercept adjusted for each country [56]. Hence, in these two cases, we used the factor mean scores adjusted after the alignment process for our planned analyses.

**Main analyses.** To test our hypotheses, we employed frequentist and Bayesian multilevel modeling (MLM) implemented in *R* packages *lme4* version 1.1–23 [57] and *brms* version 2.13.5 [58]. We standardized the measures (to M = 0, $SD$ = 1) to improve the performance of MLM [59]. As preregistered, we first performed model selection. For hypotheses 1a–b and 2a–f and for registered exploratory analyses, we examined four nested models: a model containing only an intercept (referred to as Model 0), a model with a random intercept added (Model 1), a model also containing the fixed effect(s) and interactions (if any) of interest (Model 2), and a model also containing random slopes (Model 3). For hypotheses 3a–b, we compared Models 0 and 1, as described above, to a model including a main effect of country (Model 2) and a model also including the main effect of frame and the frame by country interaction (Model 3) We selected the best model in terms of AIC, BIC, and Bayes Factors. We then examined whether, in the selected model, the effects of interest were significantly greater than zero. In addition, we calculated standardized coefficients ($\beta$) and standard errors to better understand the effect sizes of the variables of interest (See S1 File for more details).

Because the *p*-value threshold, $p < .05$, could be too liberal and lead to false positives [60, 61], we also used Bayesian MLM and Bayes Factors (BF) as additional references in our hypothesis testing. While a *p*-value is highly likely to reject the null hypothesis with a large sample, BF favors the null over the alternative hypothesis under the same condition because it penalizes unnecessarily complex models [62, 63]. Bayesian MLM was performed only in case of significant outcomes from frequentist MLM because significant outcomes are required to reject the null hypothesis [62, 64]. Whether the effect of interest was significantly greater than zero was examined with the resultant BF. We used the default Cauchy prior distribution for regression analysis, Cauchy (0, 1) [65]. Once the BF was calculated, we examined whether it exceeded the threshold for at least positive evidence supporting the alternative hypothesis, BF $\geq$ 3 [60, 66].

We applied the same analytical strategy for the exploratory analyses. Further details about how each hypothesis was tested and how the planned exploratory analyses were conducted are available in S1 File.

## Results

### Risky choice and the framing effect: COVIDiSTRESS vs. the many labs data

**Comparing stress levels with pre-pandemic estimates.** Hypotheses 1a and 1b depended on perceived distress being higher during the studied period than before the pandemic [9]. To

test this assumption, we compared our mean PSS-10 score to the best available pre-pandemic estimate of PSS-10 among non-clinical participants: the average score on PSS-10 among 2,000 Americans in 2009, a year after the economic crisis [27]. The mean score among 1,652 Americans in our study (17.34, $SD$ = 7.44) was significantly higher than the mean in 2009 (15.84, $SD$ = 7.51), $t(3,650)$ = 6.70, $p < .001$, Cohen's $d$ = 0.22, BF = 340,937,591.00. The distress level during our study was thus higher than in another stressful period marked by an economic crisis.

We could not find any studies examining the change in PSS-10 scores during the period of 2009–2019. However, the annual reports of the Stress in America survey conducted by the American Psychological Association included a one-item stress level measure that showed strong correlation with the PSS-10 [67]. According to the annual reports, the overall stress level decreased among Americans during the period of 2009–2019 [68]. Although limited to one country, these results are consistent with our assumption that the overall distress level during our study was higher than that before the pandemic. We thus proceeded to test Hypotheses 1a and 1b.

**Risky choice (Hypothesis 1a).** We predicted that the proportion of safe choices on the DP would be larger than in the Many Labs dataset [9]. We selected Model 2 (fixed effect of study + random intercepts) as the best model based on two out of three criteria (AIC and BF, see Table 3). The effect of study (Many Labs vs. our study) was significantly smaller than zero, $B$ = -.12, $SE$ = .04, 95% Bayesian Confidence Interval (CI) [-.18 -.05], β = -.03,

**Table 3. Hypothesis testing: Model comparison.**

| Hypothesis | Model | AIC | BIC | Log BF (vs. Model 0) |
|---|---|---|---|---|
| H1a | 0 | 116,289.70 | 116,299.10 | - |
|  | 1 | 115,388.90 | **115,407.50** | 449.14 |
|  | 2 | **115,382.80** | 115,410.80 | **449.81** |
|  | 3 | 115,384.70 | 115,422.00 | 447.63 |
| H1b | 0 | 122,147.50 | 122,156.90 | - |
|  | 1 | 119,406.10 | 119,424.80 | 1,369.23 |
|  | 2 | 111,059.60 | 111,087.70 | 5,538.20 |
|  | 3 | **110,862.40** | **110,899.90** | **5,635.33** |
| H2a/b | 0 | 122,147.50 | 122,156.90 | - |
|  | 1 | 121,235.70 | 121,254.40 | 454.36 |
|  | 2 | 112,754.70 | 112,801.60 | 4,683.00 |
|  | 3 | **112,563.60** | **112,638.70** | **4,770.40** |
| H2c/d | 0 | 122,147.50 | 122,156.90 | - |
|  | 1 | 121,235.70 | 121,254.40 | 454.41 |
|  | 2 | 112,760.30 | 112,807.30 | 4,680.18 |
|  | 3 | **112,573.20** | **112,648.30** | **4,765.05** |
| H2e/f | 0 | 122,147.50 | 122,156.90 | - |
|  | 1 | 121,235.70 | 121,254.40 | 454.40 |
|  | 2 | 112,763.70 | 112,810.60 | 4,678.57 |
|  | 3 | **112,531.70** | **112,606.80** | **4,787.62** |
| H3a/b | 0 | 122,147.50 | 122,156.90 | - |
|  | 1 | 121,235.70 | 121,254.40 | 454.47 |
|  | 2 | 121,181.00 | 121,631.60 | 395.71 |
|  | 3 | **112,492.70** | **113,384.50** | **4,680.46** |

*Note.* Numbers in bold represent the best model for the respective hypothesis.

standardized $SE$ = .01, $z$ = -2.84, $p$ = .003. These findings were corroborated by Bayesian MLM. The resultant BF indicating the support for our alternative hypothesis over the null was 999.00, suggesting that evidence very strongly favored Hypothesis 1a. Hence, Hypothesis 1a was supported.

**The framing effect (Hypothesis 1b).**   We predicted that the framing effect on the DP would be larger than in [9]. Model 3 (fixed effects + random slopes) was the best model in terms of AIC, BIC, and BF (Table 3). The effect of the frame was significant, $B$ = 1.24, $SE$ = .04, β = 1.24, standardized $SE$ = .04, $z$ = 29.94, $p$ < .001. We converted $B$, which is the log odds ratio, into Cohen's $d$ = .69 and compared it to the respective effect size reported by the Many Labs, $d$ = .60, using Bayesian MLM. Evidence very strongly supported that $d$ in our study was greater than that in the Many Labs, estimated difference = .08, 95% Bayesian CI [.04 .13], BF = 3,999.00. Thus, Hypothesis 1b was supported. (See, S2.1 and S2.2 Tables in S2 File for counts and proportions of safe and risky choices by frame in our study vs. [9]).

*Non-preregistered supplementary analysis on Hypothesis 1b.* Because the majority of participants in the Many Labs study [9] were from the United States (> 70%), we compared the framing effect between the two studies only among American participants, 4,610 cases in the Many Labs versus 1,652 cases in our dataset. The effect of frame on choice in our American subset was significant, $B$ = 1.37, $SE$ = .10, β = 1.37, standardized $SE$ = .11, $z$ = 13.03, $p$ < .001. The converted Cohen's $d$ was .75, which was significantly greater than in the Many Labs, Cohen's $d$ = .63, estimated difference = .12, 95% Bayesian CI [.02 .21], BF = 48.38. Hence, Hypothesis 1b was consistently supported even after controlling for potential influences from the between-country aspects.

Using Bayesian MLM, we also compared the framing effect between the two datasets for each gender individually. The framing effect among female participants was significantly larger in our study than in the Many Labs, $d$ = 0.72 vs. 0.66, BF = Infinite. However, evidence positively supported absence of such a difference among male participants, $d$ = 0.68 vs. 0.77, BF = .10.

## Feelings of distress, coronavirus concerns, and trust

**Distress, risky choices, and the framing effect (Hypotheses 2a and 2b).**   We hypothesized that higher scores on PSS-10 would be significantly associated with a larger proportion of safe choices on DP (Hypothesis 2a) and a larger framing effect (Hypothesis 2b). Model 3 (fixed effects + random slopes) was the best model in terms of all three criteria (see Table 3). The main effect of PSS-10 was non-significant, $B$ = .01, $SE$ = .01, β = .03, standardized $SE$ = .03, z = .97, $p$ = .17. Thus, H2a was not supported.

However, the PSS-10 × frame interaction was significantly greater than zero, $B$ = .03, $SE$ = .02, 95% Bayesian CI [.02 .06], β = .04, standardized $SE$ = .02, $z$ = 1.93, $p$ = .03, BF = 256.67 (see S1.1 Fig in S1 File). Hence, H2b was supported.

*Non-preregistered supplementary analyses on Hypothesis 2a and 2b.* Using MLM, we examined the relationship between distress level and safe choices in each framing condition. The proportion of safe choices was significantly positively associated with distress only in the gain frame, not in the loss frame. We also compared distress levels among participants who completed the current survey during the first vs. the second month of the survey period. There was no difference, so no habituation could be substantiated. A third additional analysis indicated that participants living in the countries with more COVID-19 deaths per million were more risk averse. However, the framing effect was not associated with COVID-19 deaths across different countries (See S2 File for further details).

**Coronavirus concerns, risky choices, and the framing effect (Hypotheses 2c and 2d).**
We hypothesized that higher levels of coronavirus concerns would be associated with a larger proportion of safe choices (Hypothesis 2c) and a larger framing effect (Hypothesis 2d). Model 3 (all fixed effects + random slopes) was the best model in terms of all three criteria (Table 3). Contrary to expectations, the main effect of the coronavirus concerns was not significantly greater than zero, $B = -.02$, $SE = .01$, $\beta = -.03$, standardized $SE = .02$, $z = -1.53$, $p = .94$. Thus, Hypothesis 2c was not supported and the direction of the association was opposite to our hypothesis.

Yet, there was a significant positive coronavirus concerns × frame interaction, $B = .05$, $SE = .01$, 95% Bayesian CI [.02 .07], $\beta = .08$, standardized $SE = .02$, $z = 3.72$, $p < .001$, BF = 124.00. In other words, larger coronavirus concerns were associated with a larger framing effect (see S1.2 Fig in S1 File). Thus, Hypothesis 2d was supported.

*Non-preregistered supplementary analysis on Hypothesis 2c and 2d.* Not only was the association between coronavirus concerns and safe choices statistically non-significant, but its negative direction was contrary to our Hypothesis 2c. To better understand the relationship, we conducted an additional exploratory analysis on the relationship between concerns and safe choices, with the frame excluded from the modeling. We undertook the same steps of model comparison as in the planned analyses (see S1.2 Table in S1 File). The main effect of coronavirus concerns was not different from zero, $B = .01$, $SE = .01$, $\beta = .02$, standardized $SE = .01$, $z = 1.69$, $p = .09$. In other words, we found no evidence that coronavirus concerns are associated with safe choices in any direction.

**Trust, risky choices, and the framing effect (Hypotheses 2e and 2f).** We hypothesized that lower trust in the country's government and health system would be associated with a larger proportion of risky choices (Hypothesis 2e) and a smaller or reversed framing effect (Hypothesis 2f). Model 3 (fixed effects + random slopes) was best in terms of all three criteria (Table 3). The main effect of trust was not significantly greater than zero, $B = .01$, $SE = .01$, $\beta = .03$, standardized $SE = .03$, $z = .91$, $p = .18$, and neither was the trust × frame interaction, $B = .02$, $SE = .02$, $\beta = .03$, standardized $SE = .03$, $z = .98$, $p = .16$ (see S1.3 Fig in S1 File). Thus, Hypotheses 2e and 2f were not supported.

## Variability of risky choices and the framing effect across countries

We hypothesized that the proportion of risky choices (Hypothesis 3a) and that the size of the framing effect (Hypothesis 3b) would vary across countries. Model 3 (fixed effects of country, frame, and country x frame + random intercepts) was the best model (Table 3). Type III Wald $\chi^2$ test indicated that both a main effect of country, $\chi^2 (46) = 515.75$, $p < .001$, and a frame × country interaction, $\chi^2 (46) = 314.51$, $p < .001$, were statistically significant. Both Hypotheses 3a and 3b were thus supported by frequentist MLM.

Bayesian MLM also supported Hypotheses 3a and 3b. We examined Hypothesis 3a by comparing Model 2, the model including the main effect of the country, and Model 0, the null model. The resultant difference in the log BF was 395.71 indicating that evidence very strongly supported the presence of the country-wise difference in risky choices. To test Hypothesis 3b, we compared Model 3 to a model omitting the interaction term, Model 2.5 (See S1 File). The difference in the log BF was 551.62, indicating that Model 3 including the frame × country interaction was significantly better.

## Exploratory analyses

**GDP per capita, risky choice, and the framing effect.** GDP per capita information (as of 2017) was acquired from the World Bank, except for Taiwan's, which was acquired from

**Table 4. Exploratory analyses: Model comparison.**

| Exploratory analysis | Model | AIC | BIC | Log BF (vs. Model 0) |
|---|---|---|---|---|
| **log(GDP per capita)** | 0 | 122,147.50 | 122,156.90 | - |
| | 1 | 121,235.70 | 121,254.40 | 454.42 |
| | 2 | 112,713.10 | 112,760.10 | 4,704.52 |
| | 3 | **112,573.20** | **112,648.30** | **4,768.12** |
| **Compliance** | 0 | 122,147.50 | 122,156.90 | - |
| | 1 | 121,235.70 | 121,254.40 | - |
| | 2 | 112,768.50 | 112,815.50 | - |
| | 3 | **112,581.10** | **112,656.20** | - |
| **Familiarity with the DP** | 0 | 122,147.50 | 122,156.90 | - |
| | 1 | 121,235.70 | 121,254.40 | 454.37 |
| | 2 | 111,567.40 | 111,633.00 | 5,273.71 |
| | 3 | **111,377.00** | **111,546.50** | **5,356.39** |

*Note.* Numbers in bold represent the best model for the respective hypothesis.

countryeconomy.com. The values of GDP per capita were standardized for better convergence in both frequentist and Bayesian MLM. Model 3 (fixed effects + random slopes) was the best model (Table 4). As prior studies have found a positive association between GDP per capita and risk aversion, we performed one-tailed tests to examine the effect of log(GDP per capita). Both frequentist and Bayesian MLM showed that log(GDP per capita) was positively associated with safe choices, $B = .06$, $SE = .02$, $\beta = .12$, standardized $SE = .05$, $z = 2.58$, $p = .005$, $BF = 99.00$. However, the association of log(GDP per capita) with the framing effect was statistically non-significant, $B = .04$, $SE = .03$, 95% Bayesian CI [.02 .11], $\beta = .06$, standardized $SE = .04$, $z = 1.42$, $p = .08$ (see S1.4 Fig in S1 File).

**Compliance, risky choice, and the framing effect.** AIC and BIC indicated that Model 3 (fixed effect + random slopes) was the best model (Table 4). The main effect of compliance was non-significant, $B = .01$, $SE = .01$, $\beta = .02$, standardized $SE = .02$, $z = .99$, $p = .16$. The compliance × frame interaction was non-significant as well, $B = .01$, $SE = .01$, $\beta = .01$, standardized $SE = .02$, $z = .35$, $p = .37$ (see S1.5 Fig in S1 File).

**Familiarity with the disease problem, risky choice, and the framing effect.** 69,370 participants were not familiar with the DP, 6,946 were familiar with the DP, and 10,949 participants were not sure. Model 3 (fixed effects + random slopes) was the best model (Table 4). Relative to the unfamiliar group, the familiar group had a higher proportion of safe choices, $B = .15$, $SE = .04$, 95% Bayesian CI [.03 .22], $\beta = .08$, standardized $SE = .02$, $z = 3.40$, $p < .001$; however, the resultant Bayes Factor, 2.44, suggests that the association could only be anecdotally supported by evidence. The familiar group also showed a lower framing effect than the unfamiliar group, as indicated by a negative familiarity × frame interaction, $B = -.28$, $SE = .06$, 95% Bayesian CI [-.39 -.15], $\beta = -.11$, standardized $SE = .02$, $z = -4.81$, $p < .001$, $BF = $ Infinite. The unsure group was not significantly different than the unfamiliar group in terms of safe choices, $B = .06$, $SE = .03$, $\beta = .04$, standardized $SE = .02$, $z = 1.63$, $p = .10$, but showed a lower framing effect, $B = -.12$, $SE = .05$, 95% Bayesian CI [-.22 -.03], $\beta = -.06$, standardized $SE = .02$, $z = -2.48$, $p = .007$, $BF = 3.03$.

## Discussion

The COVID-19 pandemic has been a major source of psychological distress [69, 70], thereby providing the opportunity to examine how distress and related emotional responses have

affected various behavioral outcomes at a global scale. We focused on how various negative emotions would impact choices in the Disease problem, which is highly evocative of the COVID-19 pandemic during which this replication took place. We further investigated cross-national variations in risk preferences and explored potential associations between the framing manipulation and compliance with safety guidelines.

## Testing the appraisal-tendency framework's predictions

Drawing on the appraisal-tendency framework (ATF) [2–4], we predicted that higher distress would be associated with a larger proportion of safe choices and a larger framing effect. Conversely, we predicted that lower trust in the government's effort to cope with the coronavirus would be associated with a smaller proportion of safe choices and a smaller framing effect. We found support for all three hypotheses relating the framing effect to distress. In particular, the framing effect was larger than in a reference study under more typical circumstances [9] (Hypothesis 1b), and was related to stress levels (2b) and concerns with the coronavirus (2d) within our study. Regarding hypotheses related to safe choices, the evidence was mixed. Although the proportion of safe choices was larger than in the reference study [9] (Hypothesis 1a), it was related to neither stress levels (2a) nor coronavirus concerns (2c) within our study. Finally, trust in the government's efforts to handle the pandemic was related to neither safe choices (2e), nor the framing effect (2f).

Our support for Hypothesis 1a is consistent with previous evidence for increased risk aversion relative to pre-pandemic levels during the February 2020 lockdown in Wuhan, China—the starting point of the COVID-19 pandemic [71]. Extending that finding, we found evidence for a global, not just local increase of risk aversion, in a period when stress levels were also higher than before. This positive but indirect evidence for an association between stress and risk aversion is qualified, however, by the non-significant results of testing the association directly within our study. The latter suggest that people's preference for safe choices on the DP has increased during the pandemic regardless of their individual level of distress or concerns.

Additional analyses ruled out the possibility of a ceiling effect in distress measures (PSS-10: global mean = 2.61, *SD* = .73, median = 2.60, min. = 1.00, max. = 5.00, skewness = .21, kurtosis = -.34, .03% of participants scored 5.00; concerns: global mean = 4.46, *SD* = 1.11, median = 4.67, min. = 1.00, max. = 6.00, skew = -.71, kurt = .10; 11.44% of participants scored 6.00). The lack of association between stress and risk avoidance was thus not due to reduced variation of stress levels. It is also possible that methods involving many tasks with varying levels of risk and trade-offs, and possibly with real rather than hypothetical outcomes [e.g. 72–74] might measure risk aversion more reliably and thus provide a better chance of finding the hypothesized association. Still, we drew our hypotheses based on studies which specifically found a positive association between distress and safe choices on the DP [2, 4]. Our null results thus cannot be explained away by the specific nature of the problem we used.

Our results diverge from previous large-scale studies with American participants [25, 26] that found that induction of fear increased perception of future risk, as predicted by the ATF. However, those studies are methodologically different from ours in potentially important ways. They have asked participants to assess risks related to their lives while we used a hypothetical framing problem. They also manipulated emotions experimentally by inducing anger or fear while we only measured the relevant variables. Given the relatively small effects from their manipulations (*d*s ranging from 0.14 to 0.30), it is possible that simply measuring the association might lead to an even smaller effect which can be diluted by the variability of our data.

An important question is whether ATF can account for the selective support regarding the framing effect but not preferences for safe choices. One possibility is that, in this study, higher levels of distress or concerns were accompanied by higher sensitivity to the context (and thus a larger framing effect) but not by appraisals of uncertainty or lack of control, which are the prerequisites for risk avoidance according to ATF. However, this explanation seems unlikely, given the strong relationship between distress, uncertainty, and lack of control [e.g., 75, 76]. Alternatively, unlike sensitivity to context, appraisals of uncertainty and lack of control might not have had a carryover effect on choices on the DP. A challenge to be addressed in future research on ATF is, therefore, to specify the necessary and sufficient conditions for the carryover to take place.

The lack of significant association between risk avoidance, the framing effect, and trust in the government's efforts to handle the pandemic may be due to several reasons. First, due to lack of measurement invariance and following [5], we only examined trust in governmental efforts, not trust in other domains we measured, such as trust in public health workers, broadcasting systems, etc. that might be more representative of the trust experienced during the pandemic.

Second, extreme and salient events affecting people's lives at a large scale apparently evoke a dominant emotion whose impact is more evident relative to other emotions [77]. Thus, anger was the dominating emotion among Americans in a study conducted shortly after the September 11th terrorist attacks, regardless of the specific emotional manipulation [25]. Similarly, distress and concerns, apparently prevailing during the period of our data collection, might have a permanent effect on people's behavior. By contrast, emotions and cognitions related to distrust might not have exerted any tangible impact on people's choices unless explicitly invoked.

Third, our hypotheses related to trust drew on the additional assumption that distrust is accompanied by cognitive appraisals similar to anger and hostility. Since we have not measured cognitive appraisals, we cannot tell whether that key premise was met. Finally, there is growing evidence that trust and distrust are two qualitatively distinct constructs rather than two poles of the same dimension [28, 30]. Testing whether ATF's predictions translate to distrust might thus require a direct measure of distrust.

In sum, the null findings related to trust/distrust might be due to limitations of this study but the lack of support for hypotheses related to risk avoidance poses a challenge to the ATF. Still, the positive association between stress, concerns, and the framing effect provides support for the ATF and, more broadly, for the role of emotions in modulating the framing effect.

## Attitudes toward risk and the framing effect across nations

Consistent with previous findings [31–33], the proportion of safe choices varied significantly across countries (Hypothesis 3a). Additional exploratory analyses showed that the proportion of safe choices was higher in countries with higher GDP per capita, similar to previous research using incentivized monetary outcomes [32, 36]. The latter finding suggests that this so-called "risk–income paradox" [32] also holds beyond monetary choices.

The size of the framing effect also varied significantly across nations (Hypothesis 3b) in spite of previous findings pointing to the contrary [9]. Our findings, based on a more diverse set of countries than [9], suggest that differences in susceptibility to framing exist on the country level which have so far been overlooked. For instance, in our study, safe choices across countries were positively associated with population-adjusted COVID-19 deaths. There might be other country-level factors, some specific to the research context, others generalizable across a variety of situations, which might also account for the variability of framing effect across

countries. Uncovering societal level factors that influence individual-level decisions is not only theoretically important but also beneficial to putting the general principles into practice while adapting them to the specific societal context.

## Risky-choice framing and compliance

Non-compliance with measures preventing the spread of coronavirus is related to risk tolerance [37] and to the propensity to put others at risk for one's own benefit [38]. We were interested if similar association will be found between lower compliance and choosing the risky option on the DP. We found no evidence for such an association, which might be due to reasons in both the nature of the DP and our compliance measure. As stated in the introduction, there may be limits to the analogy between choosing the risky option on a framing problem and acting in a risky way during a pandemic, leading to a limited predictive validity of this particular hypothetical choice on (reported) real behavior. In addition, our compliance item referred to context-general cooperative behavior rather than to specific behaviors in response to safety guidelines [see also 78]. Such general compliance might be less related to decisions under risk than is adherence to specific safety guidelines.

Compliance was also not significantly associated with the framing effect, meaning that the framing manipulation, albeit successful within the DP, did not have a detectable carryover effect on compliance. A seemingly straightforward explanation for this non-significant result is that the Disease scenario, although highly evocative of the coronavirus pandemic, does not mention it explicitly, thus impeding any tangible carryover effects on behaviors during the pandemic. Yet, studies that did explicitly use coronavirus-related scenarios in risky-choice framing tasks also failed to find evidence for carryover effects. For instance, while gain vs. loss framing did impact choosing the safe vs. risky preventive programs within the framing scenario, it was not associated with support for preventive measures to fight the coronavirus outside the scenario [79]. Another study [80] found a significant three-way interaction between frame, type of scenario (coronavirus vs. the original DP), and emotionality on compliant behavior, such that participants rating high on emotionality were more willing to comply with preventive guidelines when information was framed in terms of gains but only in the case of the coronavirus scenario. However, neither the main effect of frame nor the two-way interaction between frame and scenario significantly predicted compliance, meaning that the framing manipulation did not have an effect on compliance *regardless* of emotionality. Emotionality, on the other hand, was positively related to compliance irrespective of frame or scenario. Apparently, the framing effect is limited to the specific situation to which the framing manipulation has been applied, and does not carry over to subsequent judgments, decisions, and actions.

In the above studies, the framing manipulation did not directly target compliance behaviors. One might thus argue that, while the framing manipulation cannot prime safety choices in unrelated problems, it might still be efficient when used to directly target the behaviors of interest. Recent evidence shows that this is unlikely to be the case. For example, a study [81] did not find any impact of how the benefits are framed (i.e., 100,000 people could be saved by a lockdown vs. 100,000 people could die without a lockdown) on estimates of how long preventive measures should take place. Similarly, framing safety recommendations in terms of losses versus gains was associated with higher anxiety but not with behavioral intentions, policy attitudes, or information seeking [82]. Beyond the coronavirus pandemic, a meta-analysis has failed to find an effect of framing on compliance with health recommendation [83]. The overarching message seems to be that framing manipulations alone are not a magic tool when it comes to impacting real-world choices and behaviors: they might be effective in low-stakes

hypothetical scenarios but much less in real-life situations where the consequences of choices are much more tangible. Apparently, stable individual differences such as risk tolerance, patience, social responsibility [37], prosociality [38], and emotionality [80] have more power to drive compliant behavior than the subtle contextual manipulations provided by framing.

### Familiarity and the framing effect

Participants who reported being familiar with the DP and participants who were unsure showed less framing effect than participants who reported unfamiliar. Our decision not to exclude participants based on their familiarity with the DP might thus have lowered the overall estimate of the framing effect. Additionally, if participants from a particular country were disproportionately more familiar with the problem, their responses might bias the results and potentially the conclusions. In any case, given our results and the growing public knowledge of the framing effect through popular books [e.g., 84], researchers should consider controlling for familiarity with the particular problem they use or with the framing effect more generally.

### Limitations

Our study taps a particular period of individuals' experience of the pandemic, and some of the results might be specific to that period. We also used a snowball-sampling method for recruiting participants, meaning that our national samples are not guaranteed to be representative and have widely varying sizes. Accordingly, we could not control for a range of potentially hidden moderators (e.g., age, educational status, etc.). Furthermore, our findings are based on a single framing problem which may be viewed in a special light by respondents during a deadly epidemic and thus might not generalize to other decision-making scenarios.

Regarding our main predictions, we manipulated the frame but not participants' emotions; thus our evidence regarding the appraisal-tendency theory is mostly correlational. Furthermore, we did not measure the hypothesized cognitive appraisals that mediate the impact of specific emotions on framing. Our positive findings thus do not selectively support ATF to the exclusion of alternative accounts, and similarly our null findings do not necessarily point to weaknesses of the theory. Still, the present pattern of positive and null findings can potentially inform future work within ATF and research on the role of emotions in risky choices in general.

### Conclusion

Our findings, based on 88,181 participants from 47 countries, bear important implications for both theory and practice. Higher stress and concerns were selectively associated with the framing effect but not necessarily with the proportion of safe choices in the Disease Problem. This evidence partially supports but also challenges the appraisal-tendency framework in ways that need to be addressed in future research. Cross-country variations in risk aversion and the framing effect further suggest that important macro-level predictors of choice under risk are yet to be uncovered, with potential practical implications for large-scale crisis management. Also on the practical side, our failure to find evidence for an association between the framing manipulation and the willingness to comply with safety guidelines—along with similar failures from other studies conducted during the coronavirus pandemic [79–82]—suggests that the framing manipulation is of limited use as a tool to promote compliant behavior. Distinguishing between what framing can and cannot achieve may be of importance to both researchers and authorities around the world searching for the optimal way of communicating their messages during a crisis.

## Supporting information

**S1 File. Supplementary methods and results (with S1.1-S1.2 Tables and S1.1-S1.5 Figs).**
(PDF)

**S2 File. Analyses requested by reviewers (with S2.1-S2.2 Tables and S2.1-S2.2 Figs).**
(PDF)

## Acknowledgments

We would like to thank all the researchers involved in the COVIDiSTRESS Global Survey, and specifically to Thao Tran, Tao Coll-Martín, Dominik Ćepulić, Jesper Rasmussen, and Sabrina Stöckli, for their participation in discussions related to this paper.

## Author Contributions

**Conceptualization:** Nikolay R. Rachev, Hyemin Han, David Lacko, Rebekah Gelpí, Yuki Yamada, Andreas Lieberoth.

**Data curation:** Hyemin Han, Yuki Yamada.

**Formal analysis:** Nikolay R. Rachev, Hyemin Han.

**Funding acquisition:** Yuki Yamada.

**Investigation:** Nikolay R. Rachev, Hyemin Han, David Lacko, Rebekah Gelpí, Yuki Yamada, Andreas Lieberoth.

**Methodology:** Nikolay R. Rachev, Hyemin Han, Andreas Lieberoth.

**Project administration:** Nikolay R. Rachev, Hyemin Han, Andreas Lieberoth.

**Resources:** Hyemin Han.

**Writing – original draft:** Nikolay R. Rachev, Hyemin Han, David Lacko, Rebekah Gelpí.

**Writing – review & editing:** Nikolay R. Rachev, Hyemin Han, David Lacko, Rebekah Gelpí, Yuki Yamada, Andreas Lieberoth.

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
