## [Decision Letter · Decision Letter 0]

13 May 2021

PONE-D-21-06048

Replicating the “Disease” framing problem during the 2020 COVID-19 pandemic: A study of stress, worry, trust, and choice under risk

PLOS ONE

Dear Dr. Rachev,

Thank you for submitting your manuscript to PLOS ONE. After careful consideration, we feel that it has merit but does not fully meet PLOS ONE’s publication criteria as it currently stands. Therefore, we invite you to submit a revised version of the manuscript that addresses the points raised during the review process.

I received two high quality reports back on your paper and (as you will see in the reports) there are substantial concerns that the study does not meet the scientific criteria needed for publication in PLOSOne, with worry that such a criteria would not be met in a revision. Both reviewers did recommend the paper be given an opportunity for a revision, however.  Any attempt at a revision must satisfy the reviewers that the paper is sufficient in scientific merit (design, data analysis, etc.).  I share concerns with the reviewers regarding the paper. While the reports are very detailed I want to highlight a few additional things from my own reading.  

First and foremost the major motivation and hypotheses appear to be comparing your new data to that of data collected in a different paper in 2014. While I think it is safe to conclude/assume that distress is different in these two data sets, there is quite substantially more things that have occurred in the gap period that could plausibly affect differences as well.  I do not see a way to remove the plausibility of these other unobserved factors being responsible for observed differences. While any comparisons within your new data set do not have that problem, any discussion of differences to a pre-covid era data set would need to provide some form of additional identification mechanism. I do not see this as feasible.  That said, the data set you do have is rich, but as written the paper seems to rest too much on a comparison with no identification mechanism.  Second, you lost a lot of data to exclusions. It seems you lost about half of your data (173k down to 88k).  This seems excessive and worrisome.  How can we be confident that your remaining data is still a representative data set? Could any of the factors that contributed to individuals being excluded also be related to decision of interest behavior? Speed, completeness of answering questions, etc. could presumably be associated with dedication, honesty, concern for others, etc. that would presumably be related to variables of interest.  Your data set is also very lopsided on gender balance. Was gender balance before exclusions were implemented more or less or similarly balanced?  Does analysis look similar if you engage a weaker set of exclusion criteria? Such concerns need to be addressed. Third, touched on in the reports, is the idea of the degree of pandemic exposure differences between the different countries in the data set.  Covid had not spread as far in late March as later in the data period.  You should control for country level exposure. One reviewer suggested Covid Deaths, but you may want to explore something like Cases per 100,000 individuals or some other measure that adjusts for country population. Finally the paper is very long and cumbersome. While the topic/motivation seems concise the implementation is complicated.

When deciding whether to submit a revision, keep in mind there are no promises here. The topic and data are interesting and seem to have scientific rigor but the questions, analysis, and implementation of the study leave many concerns.

We look forward to receiving your revised manuscript.

Kind regards,

Jason Anthony Aimone

Academic Editor

PLOS ONE

Journal Requirements:

Reviewers' comments:

Reviewer's Responses to Questions

**Comments to the Author**

1. Is the manuscript technically sound, and do the data support the conclusions?

Reviewer #1: Partly

Reviewer #2: Yes

2. Has the statistical analysis been performed appropriately and rigorously? 

Reviewer #1: No

Reviewer #2: Yes

3. Have the authors made all data underlying the findings in their manuscript fully available?

Reviewer #1: Yes

Reviewer #2: Yes

4. Is the manuscript presented in an intelligible fashion and written in standard English?

Reviewer #1: No

Reviewer #2: Yes

5. Review Comments to the Author

Reviewer #1: See attached referee report for suggestions for additional data analysis and editing the paper. This main result about framing is very interesting. I suggest that the authors think about the implications of this result for public health officials, and include comments in the conclusions.

Reviewer #2: *** General comments

Generally, the paper focuses on an interesting topic, it is well-written and the statistical analyses are conducted with great care.

A main problem is that the paper focuses on many different dimensions and analyses. Overall, it is way too long and needs to be shortened.

In this respect, to increase clarity, the authors should also try to better connect and combine the different fields of the research questions they ask (e.g., the effect of

emotions on the DP paradigm and the inter-country effects could be bundled).

Belowm, I present may comments, which are ordered by section.

*** Introduction

- The authors provide interesting examples for the application to pandemic management and compliance with the healthcare guidelines. Here,

it would be important that they cite the recent literature studying these topices (Campos-Mercade et al., 2021; Müller and Rau, 2021).

- In the context of associations between risky choice and social distancing Müller and Rau (2021) also provide survey evidence of the COVID-19 pandemic.

in Müller and Rau (2021) a significantly negative relation was found between risk tolerance and compliance to healthcare guidelines.

*** Competing Accounts for the Framing Effect

- on p. 6 the Prospect Theory explanation is a bit sloppy. The authors state: "Consequently, the guaranteed gain of 200 people being saved is

evaluated as a greater gain than a one-third chance of saving 600 lives.."

=> The reason why the first gamble leads to a higher utility in the "prospect theory world" is the reference point (i.e., the sure gain of 200 people in the

first gamble). Comparing this reference point to the second gamble, it may be that people enter the loss domain, i.e., when no-one is saved,

we have a loss of -200.). In combination with loss aversion (i.e., losses loom larger than gains) the second gamble leads to a lower prospect theory

utility.

=> Similarly, in the second example, peoplke's pronounced sensitity to losses (i.e., loss aversion) is the reason why pepople in the domain of losses try

to avoid sure losses and start to gamble.

This shold be addressed, i.e., the authors should improve the explanation of the gamble choices according to the idea of Prospect Theory.

*** Predictions setting

- The paper does a good job in summarizing the literature and integrating the current study in the literature. Since, the current study compares its finding

to the "Many Labs replication" of Klein et al. (2014) it would be helpful for the reader, if this study (and its circumstances) could be better explained

and introduced. Currently, the author solely write: "..presumably run under less stressing circumstances."

- H1b: The authors write: "the framing effect on the disease problem would be higher than typical" This is too imprecise. What do the authors mean with "higher"?

The relative change in proportions of the safe choices? More detailed explanation is needed.

- I doubt that hypotheses 2a-2d are needed. As I understood it correctly, these hypotheses are related to the role of emotions on the framing effect during the pandemic. This is actually, what is also tested in hypotheses 1a and 1b.

Put differently, the reason why there should be differences as compared to Klein et al. (2014) is motivated by the changed emotions during the pandemic and their impact on frmaing (which is motivated in hypotheses 2a-2d).

I would recommend to drop hypotheses 2a-2d. Instead, hypotheses 1a and 1b should be more precisely motivated by the changes in emotions during the pandemic (see hypotheses 2a-2d).

Hypotheses 1a and 1b could be accompanied by hypotheses 2e and 2f, which could turn to hypotheses 2a) and 2b).

*** Sample

- It is conspicuous that the majority of the participants in the sample is female (72.5%). Regarding common gender differences in risk taking (i.e., women are more risk averse than men) it

is possible that this composition towards such an extreme share of women may have led to an upward bias of safe choices. In this respect, it would be important to compare and discuss the

gender composition in Klein et al. (2014). Ideally, an overview table with the sample characteristics of Klein et al. (2014) should be presented and compared to the sample of the current study.

- How do the data look like, if the authors weight it in terms of the gender composition?

*** Results

- Regarding the robustness check of the mean PSS-10 score among Americans in the current study, could it be that the US stress levels generally rose since 2009 (e.g. for different reasons)?

It would be interesting to focus compare the PSS-10 levels between 2009 and the mean of the period between 2010 and 2019. If there is no difference this would be convincing.

Regarding the test of H1a:

- It would be interesting, if the authors could report some descriptive statistics before turning to the regression analyses That is, what is the mean share of safe choices in each frame in the

current study and how did this look like in the reference study?

Regarding the test of H1b:

- If the idea of the analysis of H1b solely relies on showing a difference in the effect size, I wonder to what extent this result is driven by the difference in the power of the current study.

This should be made more clear.

Regarding the idea of the sub sample ronustness checks:

- As a robustness check (to harmonize the samples), the authors test H1b by solely focusing on the US samples. How does such a sub sample comparison look like when comparing the male and female

date across both studies?

Interpretation of the non-significant relation between safe choices and compliance:

- An explanation may be that the compliance question was very broad and did not directly contain connections to risky behavior during the crisis (i.e., staying at hoome, avoidance of large crowds,

eic). Instead, the way how the question was addressed may also refer to a more general behvior similar to cooperation, or social preferences, which is of different nature than risky decisions.

This should be addressed/discussed.

Interpretation of the increased preference for safe choices in the pandemic:

- Bu et al. (2020) find evidence for increased (temporarily) risk averion of subjects in Wuhan. They argue that this is related to their exposure to the crisis. They argue that higher exposure

to the criss (an higher levels of perceived stress) increase risk aversion. This argument of increased risk aversion (Bu et al., 2020) during the pandemic could also be used for the authors'

argument that the induced framing may be less effective, if the current distress level is already quite high.

- What do the data tell for countries with a higher exposure to the crisis? For instance, this could be proxied bny focusing on the infection or death rates of the countries.

- The authors argue that they could not infer country specific differences in the present study (I am also aware that this may be exhausting), but nevertheless I believ that this kind of

heterogeniety is really important to better explain the outcomes. Therfore, it would be important to learn more about the country-specific effects, related to the COVID-19 crisis.

- Could it be that you do not find a carryover effect of stress to preferences (in the form of risk attitudes), because you only prensent subjects to these (already) framed hypothetical contexts?

I could imagine that people would have changed their risk-taking behavior when presented to more general gambles (i.e., without this DP trade offs, where a sure option vs. a risk option exists).

The problem of such a setting is that it does not measure risk preferences based on different levels as in other risk-eliciation settings (e.g., Holt and Laury, 2002; Eckel and Grossman, 2002, Dohmen et al., 2011).

Instead, people only face a binary choice, which dramatically limits the action space.

*** References

Bu, D., Hanspal, T., Liao, Y., & Liu, Y. (2020). Risk taking during a global crisis: Evidence from wuhan. Covid Economics, 5, 106-146.

Campos-Mercade, P., Meier, A. N., Schneider, F. H., & Wengström, E. (2021). Prosociality predicts health behaviors during the COVID-19 pandemic. Journal of Public Economics, 195, 104367.

Dohmen, T., Falk, A., Huffman, D., Sunde, U., Schupp, J., & Wagner, G. G. (2011). Individual risk attitudes: Measurement, determinants, and behavioral consequences. Journal of the european economic association, 9(3), 522-550.

Eckel, C. C., & Grossman, P. J. (2002). Sex differences and statistical stereotyping in attitudes toward financial risk. Evolution and human behavior, 23(4), 281-295.

Holt, C. A., & Laury, S. K. (2002). Risk aversion and incentive effects. American economic review, 92(5), 1644-1655.

Müller, S., & Rau, H. A. (2021). Economic preferences and compliance in the social stress test of the COVID-19 crisis. Journal of Public Economics, 194, 104322.

6. PLOS authors have the option to publish the peer review history of their article (what does this mean?). If published, this will include your full peer review and any attached files.

Reviewer #1: No

Reviewer #2: No

---

## [Author Response · Author response to Decision Letter 0]

24 Jun 2021

As requested by the Editor, we responded to those comments in a separate file labeled 'Response to Reviewers'.

---

## [Decision Letter · Decision Letter 1]

25 Aug 2021

Replicating the Disease framing problem during the 2020 COVID-19 pandemic: A study of stress, worry, trust, and choice under risk

PONE-D-21-06048R1

Dear Dr. Rachev,

We’re pleased to inform you that your manuscript has been judged scientifically suitable for publication and will be formally accepted for publication once it meets all outstanding technical requirements.

Kind regards,

Jason Anthony Aimone

Academic Editor

PLOS ONE

Additional Editor Comments (optional):

Reviewers' comments:

Reviewer's Responses to Questions

**Comments to the Author**

1. If the authors have adequately addressed your comments raised in a previous round of review and you feel that this manuscript is now acceptable for publication, you may indicate that here to bypass the “Comments to the Author” section, enter your conflict of interest statement in the “Confidential to Editor” section, and submit your "Accept" recommendation.

Reviewer #1: All comments have been addressed

Reviewer #2: All comments have been addressed

2. Is the manuscript technically sound, and do the data support the conclusions?

Reviewer #1: Yes

Reviewer #2: Yes

3. Has the statistical analysis been performed appropriately and rigorously? 

Reviewer #1: Yes

Reviewer #2: Yes

4. Have the authors made all data underlying the findings in their manuscript fully available?

Reviewer #1: Yes

Reviewer #2: Yes

5. Is the manuscript presented in an intelligible fashion and written in standard English?

Reviewer #1: Yes

Reviewer #2: Yes

6. Review Comments to the Author

Reviewer #1: Thank you for your careful attention to the comments in the referee reports. The paper is now substantially improved.

Reviewer #2: Thanks for addressing most of the comments. Although, I do not always agree with all your responses, the revision substantially improved the paper and I can accept that. However, there are two remaining minor issues, which you should address:

1.) When discussing the potential relation between risk preferences and compliance, you may also mention that risky behavior in the DP may also reflect rather a state-dependent risk behavior (which is influenced by the gain/loss framed) trade off and less reflects a general risk attitude. This could explain the missing link of (general) risk attitudes and compliance, found in other studies.

2.) Overall, the paper still has to many sections. It is still very long and can be shortened. For instance, the „Familiarity and the Framing Effect“ section can be dropped. Maybe you mention these results on familarity in two short sentences in the conclusion. But there is no need for a whole section.

7. PLOS authors have the option to publish the peer review history of their article (what does this mean?). If published, this will include your full peer review and any attached files.

Reviewer #1: No

Reviewer #2: No

---

## [Editor Report · Acceptance letter]

4 Sep 2021

PONE-D-21-06048R1 

Replicating the Disease framing problem during the 2020 COVID-19 pandemic: A study of stress, worry, trust, and choice under risk 

Dear Dr. Rachev:

I'm pleased to inform you that your manuscript has been deemed suitable for publication in PLOS ONE. Congratulations! Your manuscript is now with our production department. 

Kind regards, 

on behalf of

Dr. Jason Anthony Aimone 

Academic Editor

PLOS ONE